# Associations between Orthorexia Nervosa, Body Self-Image, Nutritional Beliefs, and Behavioral Rigidity

**DOI:** 10.3390/nu14214578

**Published:** 2022-10-31

**Authors:** Marina Couceiro Elias, Daniela Lopes Gomes, Carla Cristina Paiva Paracampo

**Affiliations:** Programa de Pós Graduação em Neurociências e Comportamento, Núcleo de Teoria e Pesquisa do Comportamento, Universidade Federal do Pará (UFPA), Belém 66075-110, Brazil

**Keywords:** orthorexia nervosa, behavioral rigidity, body self-image distortion, unhealthy nutritional beliefs

## Abstract

Possible correlations between orthorexic self-reports, unhealthy nutritional beliefs, behavioral rigidity related to following rules, and distortion of body self-image were investigated. In total, 246 university students of both sexes, from different areas of knowledge, answered a sociodemographic form, the Ortho-15, the Body Shape Questionnaire, the Rigidity Scale, and a Nutritional Beliefs Form. Orthorexic self-reports were observed in 73 men and 106 women. A positive correlation was found between females and orthorexic self-reports (*p* = 0.036), and severe distortion of body self-image (*p* = 0.002) and between the latter, the behavioral rigidity scale (*p*2 = 0.189; *p* = 0.001), and female sex (*p*2 = 0.245; *p* < 0.000). In the logistic regression, women were 1.83 times more likely to present orthorexic behaviors than men. The creation of prevention and treatment strategies aimed at women is suggested and it recommended that studies investigating whether the presence of orthorexic self-reports is a risk factor for the development of eating disorders are carried out.

## 1. Introduction

The dissemination and popularization of ultra-processed foods in the media, and contradictorily the propagation of body patterns incompatible with the caloric excess of the foods disclosed exerts a significant influence on the eating behavior of individuals [1].

The media presents rules in the form of orders, advice, and suggestions for issuing certain eating behaviors, describing, as a consequence, the attainment of health and certain aesthetic standards, which can make the occurrence of these behaviors more likely [2,3] and contribute to the creation and/or propagation, depending on the period and context, of specific body patterns [4].

Eating rules implicitly and/or explicitly disclosed by the media and the individual’s experience tend to favor the development of nutritional beliefs, which according to Steinmann et al. [5], are beliefs in ideal diet patterns, generating concerns regarding the healthy effect of certain foods. Such beliefs, however, can describe reliable and unreliable relationships between the eating behavior, the context in which it occurs, and its consequences. Although there is no reliable relationship between the emission of the behavior and the promised consequences—behaving according to the nutritional belief may not promote health or the ideal body—many individuals tend to adopt eating patterns under the control of their nutritional beliefs and justifications/reasons presented by the media [2,6,7].

It is likely that the dietary patterns adopted by individuals are also influenced by the degree of behavioral rigidity, which may lead some individuals to tend to follow rules more than others due to individual differences generated by different life histories. Rehfisch [8] developed and validated a questionnaire called the Rigidity Scale, which allows classification of the responses along a continuum that goes from self-reports indicative of inflexibility to self-reports indicative of behavioral flexibility. Experimental results showed that, under certain conditions, participants previously classified as inflexible are more likely to follow rules than those classified as flexible [9,10,11].

In addition, excessive preoccupation with food, the body, weight, and body dissatisfaction that can generate body image distortions are present in pathological behaviors related to eating patterns, called eating disorders [12]. Body image is related to the perception of physical appearance (the idea of body size and weight) to the level of satisfaction (concern and anxiety related to appearance) and to situations avoided or sought by the individual caused by discomfort with their body [13], and can affect the ways of relating, social opportunities, and affective and professional life [14].

A pattern of eating behavior that is not considered an eating disorder but is characterized by an obsession with food health, food quality, and purity was recently highlighted in the literature and called orthorexia nervosa [1,15,16,17].

Orthorexic behaviors involve the excessive search for foods considered pure and healthy, avoidance of foods of specific colors, with dyes, pesticides, or food preservatives, and spending three hours or more a day planning what to eat and researching foods. As a consequence of this obsessively “healthy” eating, one can have a nutritionally unbalanced diet, resulting in restrictions, social isolation, and intolerance with those who eat differently, intensified by behaviors of teaching others what and how to eat. Individuals with orthorexic behaviors report stress, anxiety, shame, guilt, exacerbated fear, a sense of personal impurity, and negative physical sensations when they cannot rigidly maintain behaviors considered healthy [1,15,16,17].

Different instruments have been used to identify self-reports indicative of orthorexic behaviors, such as the Ortho-15, the Eating Habits Questionnaire, and the Dusseldorfer Ortorexie Scale. The Ortho-15 was the first instrument built to measure orthorexia nervosa, having been validated in several countries. Despite some recent studies [18,19,20,21] questioning the lack of clarity in the validation process of the Ortho-15, pointing out that robust standardization methods were not used and suggesting a more refined verification of the instrument, the Ortho-15 was, and still is, an instrument used to assess orthorexia nervosa in many studies [22,23,24].

Studies have been conducted with the aim of identifying the prevalence of orthorexia nervosa and testing whether there are associations between this eating pattern, body image distortions, body mass index (BMI), and sex in different populations. Some studies that applied the Ortho-15 [10,11,25,26,27] to physical education students and nutrition found self-reports indicative of orthorexic behaviors in more than 80% of the participants. However, Pontes and Souza [10] and Rodrigues [25] found no correlations between orthorexia and body mass index (BMI), and between orthorexia and body image distortion. In this same line of investigation, Brytek-Matera, Donini, Krupa, Poggiogalle, and Phillipa [28] and Plitchta, Jezewska-Zychowicz, and Gebski [29] investigated possible correlations between orthorexia, BMI, and sex in students from different courses and found no association between these variables and did not describe the prevalence of orthorexia nervosa by course. Lorenzon, Minossi, and Pegolo [30] found similar results with adults in general.

Two recent studies investigated possible associations between orthorexia nervosa and mental health [31] and between orthorexia nervosa and nutritional beliefs [5]. Strahler [5], based on data indicating that orthorexia nervosa (OrNe) has a healthy dimension (HeOr), crossed data from two prevalence surveys (N = 575) with adult women and men and observed a correlation between OrNe and worsening in mental health and between HeOr and improvement in mental health in men. They also observed that the negative effects on mental health decreased in the presence of higher levels of HeOr.

Steinmann et al. [5] evaluated whether nutritional beliefs that healthy foods relieve stress are related to orthorexia nervosa in 175 participants of both sexes who answered to a questionnaire (designed for the study) about beliefs concerning foods that relieve stress, and to the Dusseldorfer Orthorexie Scale. One of the eight study groups (8% of the sample) reported the belief that exclusively healthy foods relieve stress, with the same group having greater tendencies to orthorexia nervosa.

Among the studies found on the prevalence of orthorexia nervosa in university students from different courses [22,23,28,29], two recently carried studies stand out: Grajek et al. (2022) [23] and Guglielmetti et al. (2022) [22]. In the study by Grajek et al. (2022) [23], the Ortho-15 was used and the prevalence of orthorexia in 300 Polish students was compared. The sample was split into two groups, one pertaining to health areas and another to other areas. The results indicated that 44.5% of the students were at increased risk of orthorexia, being more frequent in the group that belonged to the health areas (63.5% vs. 25.8%). Guglielmetti et al. (2022) [22], in a study conducted in Italy, evaluated the prevalence of orthorexia in 671 students of health sciences, economic-humanities, sports sciences, and dietetics and nutrition, also using the Ortho-15. The prevalence of orthorexia was 30%, and no statistical difference in the risk of orthorexia among students from the four university courses was found, which are divergent results from that found by Grajek et al. (2022) [23], who found a more frequent risk of orthorexia nervosa in students from health areas. Together, the results of these studies point to the need to continue investigating the prevalence of orthorexia nervosa in students from different areas of knowledge, with the aim of verifying whether replication of the results of one of the two aforementioned studies could be found.

Considering that (a) individuals can adopt certain dietary patterns under the control of unreliable food beliefs; (b) individual differences can favor behavioral rigidity, which, in turn, can interfere with the dietary patterns adopted by people; (c) distortions of body image are present in pathological behaviors related to eating habits and body dissatisfaction; (d) divergent results were found between two recent studies that compared the prevalence of orthorexia nervosa among university students from different courses; and (e) in the databases researched, studies that evaluated the prevalence of orthorexia nervosa in students from different areas of knowledge in Brazil, or studies that evaluated possible associations between orthorexia nervosa and behavioral rigidity were absent, this study’s hypothesis 1 was formulated as follows: that orthorexia nervosa will be found in students from different areas of knowledge in Brazil, with no difference in the risk frequency for students in the health area. The null hypothesis was that orthorexia nervosa will not be found in students from different areas of knowledge in Brazil, and hypothesis 2 was formulated such that unreliable nutritional beliefs, behavioral rigidity, and body image distortion will be associated with orthorexia nervosa and the null hypothesis was that these associations will not exist.

Finally, considering the influence of the media on the dietary patterns of individuals, few studies on the prevalence of orthorexia nervosa in students from different areas of knowledge have been carried out so far, and that studies of this type have not been carried out in Brazil, further investigations of this nature will contribute to increasing the scope of knowledge about orthorexia nervosa, allowing an expansion of the generality of the results found in the literature and encouraging new research on the subject.

Accordingly, this study investigated whether students from different areas of knowledge present self-reports of orthorexic behaviors and whether there is an association between self-reports indicative of orthorexic behaviors, self-reports indicative of behavioral inflexibility/rigidity, self-reports indicative of unhealthy nutritional beliefs related to eating behavior, and distortion of body self-image. Additionally, we sought to verify whether there are differences between men and women in relation to the variables investigated.

## 2. Materials and Methods

### 2.1. Study Organization and Eligibility Criteria

This was a cross-sectional, descriptive, and analytical study, with a convenience sample of university students. Men and women between 18 and 30 years enrolled in different undergraduate courses in the areas of health, biological sciences, humanities, exact sciences, and languages of a public university in the north of Brazil, who read and signed the free and informed consent form (ICF), were included in the study. Psychology course students and all those who did not meet the inclusion criteria, and/or those with a disability that could impair their understanding of voluntary participation and prevent their understanding and completion of the instruments applied were excluded. The invitation to participate in the research was made orally and in person in the university’s living spaces. Data collection was carried out between 2018 and 2019.

### 2.2. Study Procedure and Research Tools

The invitation to participate in the research was made orally and in person in the university’s living spaces. Data collection was carried out between 2018 and 2019. A form was used to collect personal data (age, sex, and course) and four other instruments.

Body shape questionnaire [32]: composed of 34 questions with 6 response options (always, very often, often, sometimes, rarely, never). Classifies the body self-image pattern as no distortion, mild distortion, moderate distortion, or severe distortion.Ortho-15 [17]: composed of 15 questions with 4 alternative answers (always, often, sometimes, or never). Based on the score obtained, self-reports are classified as indicative of orthorexic behaviors or not.Rigidity Scale [8]: composed of 39 sentences with true or false answers. Assesses behavioral rigidity. Based on the score obtained, self-reports of the respondents are classified as flexible and inflexible.Nutritional Beliefs Form: Created for this research, it consists of a list of 30 sentences that indicate eating behaviors considered unhealthy by people with orthorexia, formulated by 12 nutritionists. These behaviors are related to food location, choice, type, quality and benefits of food, and self-control over food. Responses are either agree or disagree. Here, 50% or more of ‘agree’ answers are considered indicative that the sentences may indicate nutritional beliefs about unhealthy eating behaviors.

All instruments were applied individually in a single session in the following order: Ortho-15, Body Shape Questionnaire, Rigidity Scale, and Nutritional Beliefs Form. Participants were instructed to follow the instructions contained in each form and not to leave any blank answers.

### 2.3. Interpretation of the Tools Used

Body shape questionnaire [32]: scores less than 70 indicate no distortion of body self-image; scores from 70 to 90 indicate mild distortion of body self-image; scores from 91 to 110 indicate moderate distortion of body self-image; and scores above 110 indicate severe distortion of body self-image.

Ortho-15 [17]: scores less than 40 are considered self-reports of orthorexic behaviors.

Rigidity Scale [8]: scores from 0 to 11 indicate behavioral flexibility and scores from 29 to 39 indicate behavioral rigidity.

Nutritional Beliefs Form: 50% or more of ‘agree’ answers is considered indicative that the sentences may be nutritional beliefs about unhealthy eating behaviors.

### 2.4. Statistical Compilation

The database was typed, organized, and checked in the Excel program (Microsoft Office 2013 for Windows). Statistical analyses were performed using the SPSS program (Statistical Package for Social Sciences, IBM, New York, NY, USA), v. 20. For all tests performed, a significance of 5% was considered. Descriptive statistics were performed to characterize the sample. To identify a statistically significant difference between the categories of the orthorexic behavior variable (1 = presence of orthorexic behavior; 2 = absence of orthorexic behavior) and the other categorical variables of the study (image distortion: 1 = no image distortion/<70; 2 = mild distortion/from 70 to 90; 3 = moderate distortion/from 91 to 110; 4 = severe distortion/>110; behavioral rigidity: 1 = flexible/≤11; 2 = normal/from 12 to 27; 3 = inflexible/≥28; unhealthy nutritional beliefs: 1 = presence; 2 = absence; gender: 1 = male; 2 = female; course: 1 = languages; 2 = health; 3 = biological sciences; 4 = humanities; 5 = exact sciences), the chi-square test was used, with analysis of the adjusted residuals. The Spearman correlation was performed to verify the relationship between orthorexic behavior and the other variables. Prior to the logistic regression analysis, the absence of collinearity between the study variables was observed through linear regression, observing the tolerance and VIF values, all of which were greater than 0.1 and less than 10, respectively. Finally, a binomial logistic regression analysis was performed, composed of orthorexic behavior and the independent variables gender (indicator = 2/female), behavioral rigidity, and unhealthy nutritional beliefs (2 = disagreement with the assertion). The final model was able to predict 72.4% of the occurrence of orthorexic behavior in the sample studied.

## 3. Results

A total of 246 students participated in this study, 73 men and 106 women with reports indicative of orthorexia and 37 men and 30 women without reports indicative of orthorexia, with a similar distribution between genders (*p* = 0.126) and the undergraduate areas they attended (*p* = 0.225). The majority (n = 179; 72.4%) presented self-reported orthorexic behaviors, had some degree of distortion of body self-image (n = 147; 59.8%), did not present self-reports of behavioral rigidity in the extremes of flexibility and inflexibility (n = 229; 93.1%), and there were no reports of unhealthy nutritional beliefs (n = 211; 85.8%), with a statistically significant difference between these categories (*p* < 0.000) (Table 1).

A positive association was found between being male and not having self-image distortion and a positive association between being female and having severe distortion of body self-image (*p* = 0.002). In addition, a positive association was found between self-reported orthorexic behavior and being female (*p* = 0.036) (Table 2).

There was also a significant positive correlation between self-reported orthorexia nervosa and the behavioral rigidity scale (*p*2 = 0.116; *p* = 0.035). In addition, a negative correlation was found with gender (*p*2 = −0.134; *p* = 0.018), indicating that female gender was correlated with more frequent self-reports of orthorexia (Table 3).

There was a positive correlation between the presence of self-image distortion and the behavioral rigidity scale (*p*2 = 0.189; *p* = 0.001) and female sex (*p*2 = 0.245; *p* < 0.000), indicating that the degree of body image distortion is related to behavioral rigidity, which is related to the female sex. In addition, self-image distortion showed a negative correlation with the presence of unhealthy nutritional beliefs (*p*2 = −0.224; *p* < 0.000), indicating that the more intense the self-image distortion, the unhealthier the nutritional beliefs present (Table 3).

Table 4 shows the binomial logistic regression in which the dependent variable is orthorexia nervosa and the independent variables are unhealthy nutritional beliefs and gender, which showed statistical significance in the bivariate correlation. The Rigidity Scale variable was not used because only thirteen participants presented self-reports of inflexible behaviors and, among them, only eight presented self-reports of orthorexic behaviors.

Gender was a significant predictor of self-reported orthorexia nervosa, but the unhealthy nutritional beliefs variable was not. Women were 1.83 times more likely to have orthorexic behaviors than men (Table 4).

## 4. Discussion

The results found in this research indicate that most of the participants presented, according to the Ortho-15 questionnaire, presented self-reports indicative of orthorexic behaviors, regardless of gender and the area of the undergraduate course they attended. These findings replicate those obtained by Lorenzón et al. [30], who observed orthorexic self-reports in 91.4% of participants of both sexes, and those obtained by Silva and Fernandes [27], who also found orthorexic self-reports in 85.1% of participants of both sexes in both nutrition and physical education courses.

Furthermore, the results showing that Brazilian students in the areas of health, human, exact, language, and biological studies presented self-reports of orthorexic behaviors corroborate and expand the generality of the results found by Guglielmetti et al. [22], who observed a prevalence of orthorexia in 30% of the sample of 671 students from 4 different university courses, finding no statistical differences among the results of each course. On the other hand, they differ from the results obtained by Grajek et al. [23], who observed a higher risk of orthorexia in health students (63.5%) compared to students from other areas of knowledge (25.8%). It is suggested that further studies comparing the prevalence of orthorexia nervosa in students from different areas of knowledge and in the general population are carried out, with the aim of confirming the current results and those found by Guglielmetti et al. (2022) [22], which will make it possible to expand the scope of these findings.

It is noteworthy that although 72.4% of the sample, which included women and men, obtained scores indicative of orthorexia nervosa in the Ortho-15, an association was found between females and a greater presence of self-reported orthorexic behaviors, corroborating the results found by Ruiz and Quiles (2021) [7] and Ramaciotti et al. (2011) [33], who also found a higher prevalence of orthorexia nervosa in women. On the other hand, Vital et al. (2017) [11] observed a higher prevalence of orthorexia in men. Taken together, these results indicate that there is still variability in the results reported in the literature regarding the prevalence of orthorexia nervosa associated with being female or male, with some studies reporting a prevalence in females [7,33] and others in men (Vital et al. (2017)) and others without gender prevalence [19], as pointed out by Guglielmetti et al. (2022) [22] and Niedzielski and Kazmierczak-Wojtas (2021) [18].

Another associated observation was between the female sex and severe distortion of body self-image. Although there is no consensus in the literature regarding the association between gender and body self-image distortion in studies that investigated orthorexic eating patterns [10,25,30], the present results replicate those obtained by Ruiz and Quiles [7], who also observed an association between the female sex and worse perception of body image, suggesting that women can be more susceptible to issues related to eating behavior and body self-image. 

In addition, it was also observed that the more intense the self-image distortion, the greater the presence of unhealthy nutritional beliefs, a result not yet reported in the literature. Currently, there is significant pressure, emphasized by the media, to have a body image that is considered adequate by current standards. Body standards have always existed, but with the advent of social networks, people, especially women, have become more vulnerable to influences and having a body that fits the propagated standard [34]. Therefore, the findings obtained lead us to suppose that from the perception of their physical appearance and the level of body dissatisfaction, individuals can reproduce, in the form of unhealthy nutritional beliefs, rules propagated by the media about certain dietary patterns and the consequent aesthetic standards that can be achieved by adopting this dietary pattern, and will start to behave according to these nutritional beliefs, even though there is no reliable relationship between the adopted dietary pattern and the promised consequences. Direct testing of this hypothesis in future studies can lead to relevant contributions to the understanding of the interaction between body dissatisfaction, distorted self-image, inaccurate descriptions of the relationship between eating behavior and its consequences, and the adopted eating pattern.

Women are also more likely to develop eating disorders [9,34]. Although Astudillo [35] suggests that one of the differences between orthorexia nervosa and eating disorders is the concern with body image—a characteristic present in eating disorders—the findings described here show that there is an association between orthorexic behaviors and distortion of body self-image, a result supporting the hypothesis formulated in the present study. Such results indicate that the distortion of body self-image may also be present in orthorexia nervosa, bringing this type of eating behavior closer to eating disorders, which suggests that orthorexia, in addition to being female, may be a risk factor for the development of these disorders.

Another significant positive correlation observed, and until now not reported in the literature, was between self-reported orthorexic behaviors, behavioral rigidity, and being female in support of the other hypothesis formulated in this study. Behavioral rigidity is a characteristic of individuals who tend to follow rules more than others. Considering that one of the characteristics of orthorexia nervosa is the tendency to present an excessive preoccupation with food and the non-acceptance of a different diet [35,36], which involves being under the control of strict rules regarding one’s own food, it is likely that behavioral rigidity interferes with the food patterns adopted by people and is also a feature of orthorexia nervosa. Thus, based on the finding that women are more likely to present orthorexic self-reports, it seems logical that behavioral rigidity is associated with being female, which points to the need for interventions aimed at producing behavioral flexibility.

It is pertinent to say that the present study has two limitations: the fact that it is a cross-sectional study and the sample size (N = 246), which restricts the scope and generalization of the associations found, indicating the need to conduct new research, whose results may support the obtained findings. On the other hand, this study looked at new variables that have not yet been included in previous studies, such as behavioral rigidity and unhealthy nutritional beliefs. In this sense, the results found may encourage further studies to investigate associations between variables such as unhealthy nutritional beliefs, behavioral rigidity, being female, and orthorexia nervosa (new associations found in this study), deepening and expanding knowledge about this food pattern, which was recently introduced in the literature.

Finally, it can be emphasized that some recent studies [18,19,20,21] have proposed discussions about the validity of the Ortho-15, questioning the lack of clarity in the instrument validation process and arguing that there is no explanation regarding the established items and no methods of standardization. As it is an instrument that has been widely used to assess orthorexia nervosa in many studies, a more refined verification of the instrument may contribute to further support the findings obtained in research that used it as a measure of orthorexia nervosa or may contribute by pointing out the need for replication of some studies using a more accurate orthorexia nervosa assessment instrument. For example, a recent study [37] investigated the relations between orthorexia nervosa and other variables, such as eating disorders, self-esteem, and physical activity, using a different questionnaire from the Ortho-15, called the Eating Habits Questionary.

## 5. Conclusions

In summary, this study found that orthorexia nervosa is a pattern of eating behavior found in men and women and in students from different areas of knowledge, with no prevalence in any specific area. It also found that women are more likely to have this eating pattern and the severe distortion of self-image and behavioral rigidity correlated with it. It was also found that the higher the level of self-image distortion, the greater the presence of unhealthy nutritional beliefs. These results suggest the need for further studies that seek to expand the generality of the associations observed in the present study and investigate whether orthorexia nervosa can be considered a risk factor for the development of eating disorders and, thus, act in prevention.

## Figures and Tables

**Table 1 nutrients-14-04578-t001:** Descriptive characterization of the sample of students from a Brazilian Public University.

Variables		N	%	*p*-Value *
Sex	Male	111	45.1	0.126
	Female	135	54.9	
Undergraduate courses areas	Languages	52	21.1	0.225
	Health	47	19.1	
	Biological sciences	53	21.5	
	Humanities	36	14.6	
	Exact sciences	58	23.6	
Orthorexia Nervosa	With orthorexia reports	179	72.4	<0.000
	Without orthorexia reports	67	27.6	
Distorted self-image	No distortion	99	40.2	<0.000
	Mild distortion	63	25.6	
	Moderate distortion	41	16.7	
	Severe distortion	43	17.5	
Rigidity scale	Flexible	4	1.6	<0.000
	Mean	229	93.1	
	Inflexible	13	5.3	
Nutritional Beliefs	Presence	35	14.2	<0.000
	Absence	211	85.8	

* Chi-square test.

**Table 2 nutrients-14-04578-t002:** Comparison between the variables studied according to the presence or absence of reports of orthorexia by students from a Brazilian Public University.

	Sex		*p*-Value *
	Female n (%)	Male n (%)	
Distorted self-image			
No distortion	42 (17.1) ^(−)^	57 (23.2) ^(+)^	0.002
Mild distortion	34 (13.8)	29 (11.8)	
Moderate distortion	27 (11.0)	14 (5.7)	
Severe distortion	32 (13.0) ^(+)^	11 (4.5) ^(−)^	
Rigidity scale			
Flexible	2 (0.8)	2 (0.8)	0.257
Mean	123 (50.0)	106 (43.1)	
Inflexible	10 (4.1)	3 (1.2)	
Nutritional Beliefs			
Presence	19 (7.7)	16 (6.5)	0.939
Absence	116 (47.2)	95 (38.6)	
Orthorexia Nervosa			
Presence of reports	105 (42.7) ^(+)^	73 (29.7) ^(−)^	0.036
Absence of reports	30 (12.2) ^(−)^	38 (15.4) ^(+)^	

* Pearson’s chi-square test, with adjusted residual analysis; ^(+)^ Positive association ^(−)^ Negative association.

**Table 3 nutrients-14-04578-t003:** Correlation between orthorexia nervosa, self-image distortion, behavioral rigidity, and unhealthy self-rules of students from a Brazilian Public University.

		Self-Image Distortion	Rigidity Scale	Unhealthy Self-Rules	Sex
Orthorexia Nervosa	*p*2	0.034	0.116	0.138	−0.134
	*p*-value	0.300	0.035	0.070	0.018
Self-image distortion	*p*2	-	0.189	−0.224	0.245
	*p*-value	-	0.001	<0.000	<0.000

Spearman correlation test, considering *p* < 0.05.

**Table 4 nutrients-14-04578-t004:** Binomial logistic regression between orthorexia nervosa, unhealthy self-rules, and sex of students from a Brazilian Public University.

	B	S.E.	Wald	df	Sig.	Exp(B)	95% C.I. for EXP(B)	
							Lower	Upper
Nutritional Beliefs	−0.499	0.456	1.216	1	0.270	0.607	0.250	1.474
Sex	0.605	0.289	4.385	1	0.036	1.830	1.039	3.223
Constant	−1.191	0.213	31.233	1	0.000	0.304		

Binomial logistic regression, considering *p* < 0.05.

## Data Availability

The data of the present study can be obtained through correspondence with the indicated author.

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
