# Peer review of "Associations between Orthorexia Nervosa, Body Self-Image, Nutritional Beliefs, and Behavioral Rigidity"

_nutrients, 2022, doi:10.3390/nu14214578_

Round 1

Reviewer 1 Report

The paper would benefit from a more thorough review of literature, a critical understanding of the instruments used (e.g. ORTHO 15), and extensive revision of the use of the English language.

There are two substantial concerns with this article:

1).  It is not clear that this study has anything to add to the existing literature on orthorexia nervosa; it discovers some of the same correlations and patterns of behavior already posited by existing literature

2).  The prose is often nearly incomprehensible and in need of extensive editing. There are other issues; for example, there is no critical understanding of the ORTHO-15 questionnaire, which has serious flaws that are not acknowledged; the literature review seems to have some important gaps (some of which would have actually helped the author's point, e.g. Koven and Abry, 2015); and some assertions are simply not supported, such as that most studies of ON to date were conducted on "participants in the health areas" (sic), which is simply not true.

Author Response

COVER LETTER – REVIEW 1

Paper: Associations between orthorexia nervosa, body self-image, nutritional beliefs, and behavioral rigidity

ID: nutrients-1920127

We appreciate all the reviewer’s valuable suggestions. We managed to include in the text new arguments and information according to the suggestions and requests, as explained in detail in the answers that follow. We believe that the new version of the manuscript is more clear and more complete.

# Review 1:

1) The paper would benefit from a more thorough review of literature, a critical understanding of the instruments used (e.g. ORTHO 15), and extensive revision of the use of the English language.

Answers: We extended the review of the recent literature and incorporated it into the introduction, including the few studies that also compared the prevalence of orthorexia nervosa among students from courses in different areas (Grajek et al., 2022, Guglielmetti et al., 2022). Different instruments used to evaluate the orthorexia nervosa were presented, and highligted, in the introduction and in the discussion, the questions concerning the Orto-15. Additionally, the variables included and analyzed in the present study, that had not yet been tested, were highlighted in the introduction and discussion thus improving the paper’s justification.

2) It is not clear that this study has anything to add to the existing literature on orthorexia nervosa; it discovers some of the same correlations and patterns of behavior already posited by existing literature

Answers: We highlight, in the introduction and in the discussion, the variables that the present study used in its analysis that had not yet been tested in other studies, such as the relationship between behavioral rigidity, nutritional beliefs, and orthorexia nervosa.

3) The prose is often nearly incomprehensible and in need of extensive editing. There are other issues; for example, there is no critical understanding of the ORTHO-15 questionnaire, which has serious flaws that are not acknowledged; the literature review seems to have some important gaps (some of which would have actually helped the author's point, e.g. Koven and Abry, 2015); and some assertions are simply not supported, such as that most studies of ON to date were conducted on "participants in the health areas" (sic), which is simply not true.

Answers: We carried out a review of the recent literature and incorporated it into the introduction, including the two recent studies that also compared the prevalence of orthorexia nervosa among students from courses in different areas (Grajek et al., 2022; Guglielmetti et al., 2022). We included a critical review of the limitations of the use of the Orto-15 in the introduction. It is worth pointing out that the Orto-15 continues to be used in recently published studies, including this journal Nutrients (Grajek et al., 2022; Guglielmetti et al., 2022). Finally, we agree that the affirmation concerning the restriction of participants to courses from health areas was outdated and was deleted.

Reviewer 2 Report

Dear Authors,

Thank you for the opportunity to review this manuscript.

Since orthorexia was first described less than 20 years ago, there are not many large studies on its prevalence, and those available often vary widely in their results depending on the population studied and the diagnostic criteria adopted, a fact that would be worth noting in this study. Researchers dispute the place of orthorexia among mental disorders. Some argue that it is a separate diagnostic category, others note many elements linking orthorexia with anorexia, while still others classify the syndrome as a variant of obsessive-compulsive disorder, due to the phenotypic similarities between the two disorders.

The introduction, although it appears to be written correctly, could be improved, with smoother transitions between paragraphs. It would be worthwhile to add a couple of concluding sentences, rather than describing each one: 'media influence on eating behavior'; 'body image'; 'orthorexia'. There is also a lack of epidemiological data on the scale of the phenomenon worldwide and in the country where the study was conducted.

The study's sample is small considering the size of the population and the fact that the survey was implemented in different majors. The researchers found that as many as 179 (72.4%) people exhibited orthorectic behavior, this topic should be expanded on what basis this was found.

The discussion is conducted correctly with reference to the relevant literature. However, I would like to point out that a number of valuable items have recently been published in the journal Nutrients using the ORTO-15 questionnaire that would be worth citing.

I would also suggest citing by order rather than alphabetically, and adjusting the literature notation according to MDPI editorial requirements.

Orthorexia is a new and life-threatening disorder, part of the cultural phenomena of our time, requiring detailed research and the development of effective therapeutic methods based on it. Since the thoughts and behaviors of orthorexics are egosyntonic, it is difficult to convince the patient that excessive fixation on health is detrimental to him, especially in the context of ubiquitous slogans about the need to eat healthy and lead a healthy lifestyle, so the researchers have taken up an important topic that should be followed up on a larger population.

Author Response

COVER LETTER – REVIEW 2

Paper: Associations between orthorexia nervosa, body self-image, nutritional beliefs, and behavioral rigidity

ID: nutrients-1920127

# Review 2:

We appreciate all the reviewer’s valuable suggestions. We managed to include in the text new arguments and information according to the suggestions and requests, as explained in detail in the answers that follow. We believe that the new version of the manuscript is more clear and more complete.

1) Dear Authors, thank you for the opportunity to review this manuscript.

Since orthorexia was first described less than 20 years ago, there are not many large studies on its prevalence, and those available often vary widely in their results depending on the population studied and the diagnostic criteria adopted, a fact that would be worth noting in this study. Researchers dispute the place of orthorexia among mental disorders. Some argue that it is a separate diagnostic category, others note many elements linking orthorexia with anorexia, while still others classify the syndrome as a variant of obsessive-compulsive disorder, due to the phenotypic similarities between the two disorders.

Answers: In the introduction, we pointed out the small number of studies concerning the prevalence of orthorexia nervosa and in the discussion, we acknowledge the variability in the results reported in different studies, linked to the differences in the population.

2) The introduction, although it appears to be written correctly, could be improved, with smoother transitions between paragraphs. It would be worthwhile to add a couple of concluding sentences, rather than describing each one: 'media influence on eating behavior'; 'body image'; 'orthorexia'. There is also a lack of epidemiological data on the scale of the phenomenon worldwide and in the country where the study was conducted.

Answers: We carried out an extensive review of the recent literature and incorporated it into the introduction, including the few studies that also compared the prevalence of orthorexia nervosa among students from courses in different areas, however, no recent studies were found that described the prevalence of orthorexia nervosa in students in Brazil. For this reason, we describe data from similar studies found in other countries (Grajek et al., 2022 and Guglielmetti et al., 2022). We also tweaked the introduction, with smoother transitions between paragraphs, as requested.

3) The study's sample is small considering the size of the population and the fact that the survey was implemented in different majors. The researchers found that as many as 179 (72.4%) people exhibited orthorectic behavior, this topic should be expanded on what basis this was found.

Answers: We included a discussion on the limitations of the small sample studied, encouraging future studies to use a representative sample. We expanded the discussion on the prevalence of orthorexia nervosa found.

4) The discussion is conducted correctly with reference to the relevant literature. However, I would like to point out that a number of valuable items have recently been published in the journal Nutrients using the ORTO-15 questionnaire that would be worth citing.

Answer: We have included two recent studies published in this journal that also compared the prevalence of orthorexia nervosa among students from courses in different areas using the ORTO-15 (Grajek et al., 2022; Guglielmetti et al., 2022).

5) I would also suggest citing by order rather than alphabetically, and adjusting the literature notation according to MDPI editorial requirements.

Answer: We appreciate the alert and we’ve adjusted the references in citation order.

6) Orthorexia is a new and life-threatening disorder, part of the cultural phenomena of our time, requiring detailed research and the development of effective therapeutic methods based on it. Since the thoughts and behaviors of orthorexics are egosyntonic, it is difficult to convince the patient that excessive fixation on health is detrimental to him, especially in the context of ubiquitous slogans about the need to eat healthy and lead a healthy lifestyle, so the researchers have taken up an important topic that should be followed up on a larger population.

Answer: Thanks for the comment, we have included suggestions for future studies in the discussion.

Round 2

Reviewer 1 Report

The main news here is that they find more prevalence of ON in women, something that would require a little more investigation/explanation in the context of other ON studies.  

The version I downloaded it still maintains, for example, that "studies indicate the prevalence of orthorexia nervosa in students in the health area and that, to date, few studies have been conducted comparing the prevalence among university students from different courses" - which, again, not sure it's true, and also it's not established why students are or should be the major focus of the study of ON (other than that they present a convenient sample). (I guess the authors assume that health science students could be more health conscious and therefore more susceptible to ON, but they never explicitly say so.)

I am not sure the results really indicate that "orthorexia may become a public health issue" as the authors suggest - this seems like a hasty generalization that requires a higher burden of proof than the one offered here.

The language is still in significant need of editing - I would enlist the services of an English language editor, as the article is not fit for publication in its current shape.

Overall, I feel this paper has some merit but it still needs significant editing and rethinking of the conclusions and implications. 

Author Response

COVER LETTER – REVIEWER 1

Paper: Associations between orthorexia nervosa, body self-image, nutritional beliefs, and behavioral rigidity

ID: nutrients-1920127

We managed the manuscript to include in the text new changes concerning the suggestions and requests to the previous version, as explained in detail in the answers that follow.

  1. The main news here is that they find more prevalence of ON in women, something that would require a little more investigation/explanation in the context of other ON studies.

Answer: We included in the discussion results of studies indicating that there is still variability in the results reported in the literature regarding the prevalence of orthorexia nervosa associated with being female or male.

  1. The version I downloaded still maintains, for example, that "studies indicate the prevalence of orthorexia nervosa in students in the health area and that, to date, few studies have been conducted comparing the prevalence among university students from different courses" - which, again, not sure it is true, and also it is not established why students are or should be the major focus of the study of ON (other than that they present a convenient sample). (I guess the authors assume that health science students could be more health conscious and therefore more susceptible to ON, but they never explicitly say so.)

Answer: We included in the introduction and discussion the results of recently published studies (Guglielmetti et al., 2022; Grajek et al., 2022) which present divergent results regarding the prevalence of orthorexia nervosa in students from different areas of knowledge. Grajek et al. (2022) found a more frequent risk of orthorexia nervosa in health students, a result that differs from that found by Guglielmetti et al., 2022. We indicate the need to continue investigating the prevalence of orthorexia nervosa in students from different areas of knowledge, especially in Brazil, considering that in the databases researched, no studies were found that evaluated the prevalence of orthorexia nervosa in students from different areas of knowledge in Brazil.

  1. I am not sure the results really indicate that "orthorexia may become a public health issue" as the authors suggest - this seems like a hasty generalization that requires a higher burden of proof than the one offered here.

Answer: We eliminated that statement from the manuscript.

  1. The language is still in significant need of editing - I would enlist the services of an English language editor, as the article is not fit for publication in its current shape.

Answer: We guarantee that after the eventual acceptance of the manuscript we will hire the services of a professional English language reviewer.

  1. Overall, I feel this paper has some merit but it still needs significant editing and rethinking of the conclusions and implications.

Answer: We made new modifications to the manuscript conclusion, according to the new suggestions and requests.

Sincerely yours

Reviewer 2 Report

Dear Authors,

Thank you for making important changes that will certainly enhance the value of the study, while I have a few more comments:

It would be useful to have research hypotheses for the purpose.

The material and method chapter should be divided into the following sections and better described, especially the inclusion criteria and interpretation of the tools used:

2.1.Study organization and eligibility criteria

2.2.Study procedure and research tool

2.3.Interpretation of the tools used

2.4.Statistical compilation

I also do not see the significant changes regarding the described results that I requested.

Please respond to the comments point by point. 

Greetings!

Author Response

COVER LETTER – REVIEWER 2

Paper: Associations between orthorexia nervosa, body self-image, nutritional beliefs, and behavioral rigidity

ID: nutrients-1920127

We managed the manuscript to include in the text new changes according to the new suggestions and requests to the previous version, as explained in detail in the answers that follow.

  1. It would be useful to have research hypotheses for the purpose.

Answer: We managed to highlight the research hypothesis in the introduction.

  1. The material and method chapter should be divided into the following sections and better described, especially the inclusion criteria and interpretation of the tools used:

2.1. Study organization and eligibility criteria

2.2. Study procedure and research tool

2.3. Interpretation of the tools used

2.4. Statistical compilation

Answer: Material and Method were divided in sections as indicated, and an interpretation of the tools used was inserted.

  1. I also do not see the significant changes regarding the described results that I requested.

Answer: We are sorry because in the previous review we did not receive such request. Therefore we are not aware of what could be the significant requested changes.

Sincerely yours.